# Procalcitonin as a Potential Biomarker in the Study of Babesiosis Caused by *B. microti*

**DOI:** 10.3390/pathogens11111222

**Published:** 2022-10-23

**Authors:** Michael Lum, Caitlin Gauvin, Sophia K. Pham, Aikaterini Papamanoli, Eric D. Spitzer, Andreas P. Kalogeropoulos, Luis A. Marcos

**Affiliations:** 1Division of Infectious Diseases, Department of Medicine, Stony Brook University, Stony Brook, NY 11794, USA; 2Division of General, Geriatric and Hospital Medicine, Department of Medicine, Stony Brook University, Stony Brook, NY 11794, USA; 3Division of Internal Medicine, Department of Medicine, Stony Brook Southampton Hospital, Southampton, NY 11968, USA; 4Department of Pathology, Stony Brook University, Stony Brook, NY 11794, USA; 5Division of Cardiology, Department of Medicine, Stony Brook University, Stony Brook, NY 11794, USA

**Keywords:** parasitology, babesiosis, tick borne illness, biomarker, procalcitonin

## Abstract

Procalcitonin is gaining momentum in the study of protozoal sepsis, but its utility as a biomarker has yet to be fully discovered in human babesiosis. A total of 33 cases of acute babesiosis dating between 2012 and 2019 were retrospectively collected from Stony Brook University Hospital (SBUH) and Stony Brook South Hampton Hospital (SHH), both of which are located on Long Island, NY. Cases were cross-referenced for the need for ICU admission, and the procalcitonin levels were measured by the use of BRAHMS Elecsys assay at SBUH and BRAHMS Architect assay at SHH. Our study demonstrated that the log-transformed procalcitonin levels had a linear correlation with log-transformed maximum parasitemia, which suggests that procalcitonin directly correlates with parasitemia values. Furthermore, when comparing values that predict ICU admission, our ROC analysis of procalcitonin demonstrated similar AUC values to the percentage of parasitemia, suggesting that procalcitonin may assist in determining the severity of disease. We demonstrate that procalcitonin may directly correlate with the parasitemia percentage and have prognostic capabilities, which suggests that procalcitonin may have biomarker potential in human babesiosis.

## 1. Introduction

The use of biomarkers in infectious diseases can enhance the clinician’s ability to provide information regarding the diagnosis, progression, and prognosis of an infectious process. While there are many biomarkers that help guide clinicians during bacterial infections, the discovery of reliable values that can track protozoal infections has been challenging due to the unique pathophysiologic mechanisms of most parasites.

*Babesia microti*, which is the parasite responsible for the majority of human babesiosis in Northeastern US, has a unique lifecycle that allows it to invade and destroy erythrocytes, leading to the development of acute febrile hemolytic anemia. While the clinical severity can vary depending on the integrity of host immunity, severe disease can induce multi-system organ failure and lead to the development of acute hypoxic respiratory failure, congestive heart failure, disseminated intravascular coagulation, and renal failure [1]. Furthermore, it has been documented in several case reports that the misdiagnosis or delay in appropriate treatment may prove fatal in cases of severe disease [2]. Considering this, further study is required to find novel methods for the early diagnosis and recognition of severe illness in order to prevent poor outcomes. 

Procalcitonin has been gaining momentum as a potential biomarker for the study of parasitic infections. In a study of 72 dogs infected with *Babesia canis*, dogs with canine babesiosis were found to have a statistically significant elevation in procalcitonin compared to healthy dogs [3]. Multiple cases of human babesiosis have reported elevated procalcitonin levels during active infection, and these values assisted clinicians in considering an ongoing infectious/inflammatory process while trying to establish the diagnosis of babesiosis [4,5,6]. Mareedu et al. demonstrated that in 12 patients with acute babesiosis, all were found to have elevated levels of procalcitonin [7]. Elevated levels of procalcitonin have also been observed in cases of babesiosis in Europe caused by *B. divergens* [8]. The multiple reports of elevated procalcitonin during animal and human babesiosis poses the question of whether this test has potential diagnostic and prognostic potential in human babesiosis.

Given that high percentages of parasitemia are associated with severe illness during *B. microti* infection, the primary goal of this study is to determine the relationship between procalcitonin levels and parasitemia and to elucidate the potential for procalcitonin to be used as a biomarker in the study of babesiosis [1,9].

## 2. Methods

Babesia cases were identified from Stony Brook University Hospital (SBUH) and Stony Brook South Hampton Hospital (SHH) electronic health record systems from 2012 to 2019. Cases were included if patients had symptoms concerning babesiosis, procalcitonin measured during their clinical encounter and parasites detected by peripheral blood smear with Giemsa stain and later confirmed by *B. microti* DNA RT-PCR performed by the New York State Department of Health. Maximum parasitemia, which was defined as the highest percentage of parasites detected on a peripheral blood smear and usually reflecting the initial parasitemia value drawn, were recorded and median values were calculated. Similarly, procalcitonin values were ordered by the treating clinical team, almost always within 24–48 h of initial presentation. These values were recorded and the median values were calculated. The procalcitonin levels were measured by the use of BRAHMS Elecsys assay at SBUH and BRAHMS Architect assay at SHH. Procalcitonin levels less than 0.10 and 0.50 ng/mL were considered normal for Elecsys and Architect assays, respectively. Cases were then cross-referenced for the presence of severe disease, which was defined as the need for admission to the intensive care unit (ICU).

The correlation between maximum parasitemia and procalcitonin values was examined with the nonparametric Spearman rank correlation, and fractional polynomials were used to assess the functional form of the correlation (i.e., linear vs. non-linear). Receiver-operator characteristic (ROC) curve analyses were used to identify cutoff points for procalcitonin and maximum parasitemia as markers for the prediction of ICU admission. Analyses were performed with STATA 16.1 (StataCorp, College Station, TX, USA) and *p* values ≤ 0.05 were considered statistically significant.

## 3. Results

A total of 33 patients met the inclusion criteria for acute babesiosis, and the cohort’s age, gender, demographics, past medical history, and admission details were recorded (Table 1). The maximum parasitemia (range: 0.1–16.2%) and procalcitonin levels (range: 0.09–10.5 ng/mL) demonstrated a positive correlation, ρ= 0.567 (*p* = 0.0006). In fractional polynomial analysis, the log-transformed procalcitonin levels had a linear correlation with log-transformed maximum parasitemia, r = 0.556 (*p* = 0.001, Figure 1A).

Out of 33 patients, 8 required ICU admission, which was approximately 24% of the study population. Additionally, of the remaining 25 patients not admitted to the ICU, four were discharged from the ER while the remaining 21 were admitted to the hospital (Table 1). In ROC curve analysis, a cut-off level of ≥1.2 ng/mL for procalcitonin had optimal prediction characteristics for ICU admission (sensitivity 62.5%, specificity 88%; correct classification 82%) with a clinically relevant area under the curve (AUC) of 0.77 (95% CI: 0.58–0.89; *p* = 0.005, Figure 1B). Furthermore, in ROC curve analysis, a cut-off level of ≥3.6% for maximum parasitemia had optimal prediction characteristics for ICU admission (sensitivity 75%, specificity 92%, correct classification 88%) with a clinically relevant AUC of 0.75 (95% CI 0.58—0.89; *p*= 0.005, Figure 1B). Comparison of AUC for procalcitonin and parasitemia yielded no significant difference (*p* = 0.81).

## 4. Discussion

Our study found elevated procalcitonin levels in acute human babesiosis, suggesting that procalcitonin might be a potential biomarker in the study of this parasitic infection. Our analysis demonstrated a positive correlation between maximum parasitemia and procalcitonin levels (Figure 1A). In the current literature, the percentage of parasitemia is the gold standard for recognizing severe disease [1,2], as it has been demonstrated that parasitemia values > 4% were associated with a relative risk of 2.32 (0.52–10.0) for severe disease, defined as death during hospitalization, admission >14 days, or ICU admission > 2 days [9]. Our study supports this finding as parasitemia ≥ 3.6% suggested severe disease and need for ICU admission with a sensitivity 75% and specificity of 92%. Furthermore, our ROC analysis of the procalcitonin levels demonstrated similar AUC values to those of percentage of parasitemia. The procalcitonin values ≥ 1.2 ng/mL predicted the need for ICU admission with a sensitivity of 62.5% and specificity of 88% (Figure 1B). Our data suggest that measuring procalcitonin may provide adjunctive value to monitoring parasitemia levels during babesia infections.

While early studies suggested that only bacterial pathogens cause a rise in procalcitonin levels, non-bacterial processes such as fungi and certain parasites are now recognized as other causes that can increase this biomarker [10]. Severe *Plasmodium falciparum* malaria, which shares many phylogenetic and pathophysiologic features with babesiosis, demonstrated significant elevation in the procalcitonin levels during severe infection, and the procalcitonin values showed a strong correlation with the levels of parasitemia [11]. While the specific mechanism by which protozoal sepsis induces procalcitonin elevation is yet to be fully understood, *P. falciparum* infection is known to cause the excessive production of TNF-α, and the current literature suggests that cytokines such as TNF-α can stimulate extrathyroidal expression of the CALC-1 gene, leading to procalcitonin [11,12,13]. Furthermore, this cytokine has been demonstrated to play a major role in some of the most devastating complications of malaria such as cerebral malaria, severe anemia, renal failure, and ARDS [13]. Similarly, TNF-α is known to be a key cytokine released by macrophages during the innate immune response to the early intra-erythryocytic phases of babesiosis, which may help explain the positive correlation between parasitemia and procalcitonin levels and the trend toward higher levels of both biomarkers during severe disease [14].

Recognition of human babesiosis as a cause of procalcitonin elevation has several clinical implications. First, our study suggests that procalcitonin may add adjunctive value in determining the severity of illness when it is measured in tandem with the current gold standard of parasitemia percentage [1,9]. Secondly, procalcitonin is gaining momentum in the study of fever of unknown origin, as its use has been studied in febrile oncologic patients as well as those with rheumatologic conditions [15,16]. Tick-borne illness remains an important differential diagnosis in the evaluation of fever of unknown origin, and such entities may be missed if a high index of suspicion is not maintained [17]. There are few case reports of FUO due to babesiosis [18,19]. In the case of FUO due to babesiosis published by Cunha et al., the patient experienced 6 weeks of fevers and underwent outpatient workup without obtaining a diagnosis. While he was later admitted for presumed babesiosis, it was reported that it was not until day 13 that his peripheral blood smear demonstrated 3% parasitemia [19]. While procalcitonin was not measured in these cases, with further study, procalcitonin measurement may aid in similar cases of FUO, where an elevated value may broaden the differential to include parasitic infections such as subacute babesiosis in endemic areas, and the corresponding diagnostic tests can be ordered in a timely manner to prevent further complications.

There is a growing body of evidence that suggests that the incidence of tick-borne infections is increasing outside of traditional endemic areas. *B. divergens* has now been reported in many countries across Europe [8]. Similarly, the range of *B. microti* is expanding outside the northeastern and upper-midwest regions of United States into adjacent states. Due to these geographic changes, babesiosis may not always be considered in the differential diagnosis for acute febrile patients. Transfusion-transmitted babesiosis can also be overlooked when it occurs outside of a traditional endemic region. Haselbarth et al. suggest that hemolytic anemia accompanied by a positive direct antiglobulin test or an elevated procalcitonin should prompt an investigation for babesiosis [4,8]. Similarly, our study and that of Mareedu et al. support the concept that elevated procalcitonin in a febrile patient should raise the possibility of babesiosis. Mareedu et al. also observed elevated levels of procalcitonin in cases of babesiosis in Wisconsin; however, this analyte was only measured in 12 out of 128 cases, so it was not possible to determine potential correlations with the clinical outcome. Our findings complement this study and provide additional data on procalcitonin’s possible diagnostic and prognostic adjunctive potential in a larger sample size [7]. Since automated hematology analyzers may not detect intraerythrocytic parasites such as *Babesia* spp. or *P. falciparum* if a high index of suspicion is not maintained, there may be a delay in diagnosis [8,20,21]. Our findings suggest that persistently elevated procalcitonin may help clinicians outside of traditional endemic areas consider human babesiosis in their differential diagnosis, thus enabling earlier diagnosis and the prevention of adverse outcomes. 

Our study has several limitations. First, our sample size was small and cases were drawn from a single tertiary care system in a single geographic location on Long Island, NY, USA. While cases were screened for infections other than babesiosis, our study did not measure baseline procalcitonin levels in subjects prior to or after the infection.

Our results suggest that procalcitonin correlates with parasitemia and supports the notion that babesiosis may join malaria as another apicomplexan that can significantly raise the procalcitonin levels. Further study is required to elucidate procalcitonin’s full potential as a biomarker during *B. microti* infection.

## Figures and Tables

**Figure 1 pathogens-11-01222-f001:**
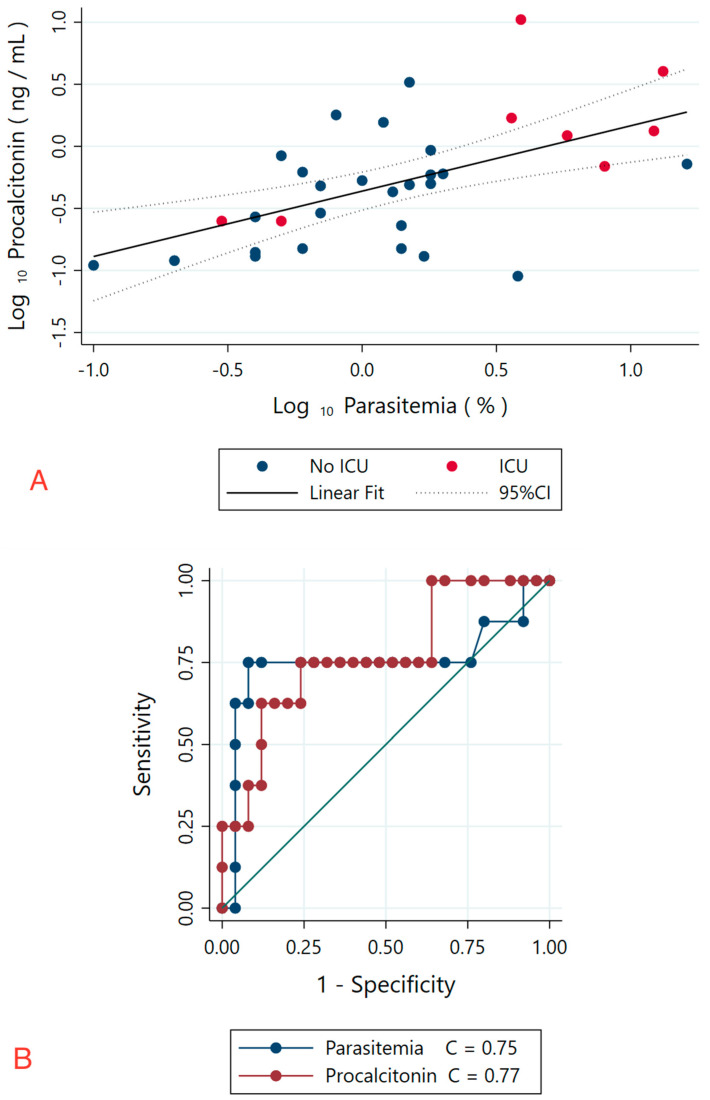
(**A**) Linear correlation between log_10_-parasitemia and log_10_- procalcitonin levels. The linear correlation of the log-transformed variables was r = 0.556 (*p* = 0.001), similar to the nonparametric (Spearman) correlation coefficient (ρ = 0.567). The red dots represent patients who were admitted to the intensive care unit. (**B**) Receiver-operator characteristic curve of parasitemia for prediction of intensive care unit admission (0.75; 95% CI: 0.58–0.89; *p* = 0.005); optimal cut-off is 3.6%. Receiver-operator characteristic curve of procalcitonin for prediction of intensive care unit admission (C = 0.77; 95% CI: 0.58–0.89; *p* = 0.005); optimal cut-off is 1.2 ng/mL.

**Table 1 pathogens-11-01222-t001:** Demographics and characteristics of patients with acute babesiosis who had procalcitonin values measured.

Patient Characteristics (Total *n* = 33)	Number of Cases	% of Cohort
Age (years)		
40–65	14	42.4%
>65	19	57.6%
Gender		
Male	21	63.6%
Female	12	36.4%
Ethnicity:		
Caucasian	19	57.6%
Hispanic	13	39.4%
Asian	1	3.0%
Comorbidities		
Immunocompromised	8	24.2%
Splenectomy	1	3.0%
Hypertension	13	39.4%
Diabetes	7	21.2%
Cardiac disease	8	24.2%
Cancer	4	12.1%
Chronic kidney disease	2	6.1%
Chronic lung disease	6	18.2%
Liver disease	1	3.0%
Autoimmune disease	3	9.1%
Admission Status		
Inpatient admission	29	87.9%
ICU admission	8	24.2%

## Data Availability

Data presented in this study can be found within the Appendix A of this article.

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
