# Peer review of "Procalcitonin as a Potential Biomarker in the Study of Babesiosis Caused by B. microti"

_pathogens, 2022, doi:10.3390/pathogens11111222_

Round 1

Reviewer 1 Report

This is a nice paper on procalcitonine an B. microti infection in humans. Critique is as follows:

1. There is a total neglegt of previous casusistic literature on the topic especially when it comes to the use of procalcitonine in European cases of human babesiosis (see below!) alrerady more ten years ago. Also American authors have casuistically reported elevated PCT levels in babesiosis patients (Krause et al, "Management of babesiosis"). This is inacceptable and suggests poor knowledge of the literature on this topic.

2. I suggest to give these observations some credit and include them in the introduction or discussion section also to extrapolate the findings to a broader diagnostic level for other babesia!

3. The combination of a positive combs test with elevated PCT levels will strengthen the positive predictive value of testing. Maybe its worth checking your patients for the presence of such a positive test too.

4. Study title: In doing so the study title would be also more suitable as the authors currently can just refere to B. microti babesiosis but tend to use the broder term human babesiosis when discussion their findings.

Some more literature to the topic:

a. Hildebrandt et al in 2021, Pathogens : Quote: "Clinical laboratory diagnosis of human babesiosis is challenging and it is uncertain whether automated hematology analyzers can reliably detect piroplasms. Where there are typical clinical symptoms, a positive Coombs test in combination with hemolytic anemia and elevated procalcitonine levels is highly suggestive of babesiosis and should promptfurther diagnostic testing."

b. Even earlier Hunfeld et al in 2008 in  the J. of Parasitology already report the suitability of PCT for cases in Europe both B. venatorum and B. microti cases:  Quote: "A positive Coombs test in combination with hemolytic anemia and elevated procalcitonine levels is highly suspicious of babesiosis (Häselbarth et al., EJMMID, 2007 (case of B. venatorum]; Hildebrandt et al., INFECTION; 2007 [case of B. microti]) and should prompt further diagnostic tests."

Again, it is the combination of PCT and a positive Coombs test wich paves the way for possible laboratory diagnosis of human babesiosis by prompting a blood smear and/or PCR for babesia.

The same is true for Malaria. In both infections the amount od PCT elevation tends to correlate with the  burden of the parasite and the severity of the disease as also demonstrated and discussed by the authors!

Reviewer 2 Report

Dear authors,

Congratulation for this study. Although I find this topic of high interest, I have several concerns:

1. How did get the procalcitonin values? Were they measured routinely during patient hosposlisation or you preformed analysis for the purpose of this research?

2. Why haven't you specified the dates (or day from illness onset) when comparing your findings of procalcitonin level and parasitemia. It is important to know if the blood for given analyses was sampld on the same day or no. In addition, was the sampling performed at the time of hospitalization or anytime later?

Please check the Latin names of mentioned microorganisms. 

Round 2

Reviewer 1 Report

No additional comments

Author Response

Thank you for taking the time to review our manuscript. Your input was greatly appreciated!

Reviewer 2 Report

Dear authors,

Thank you for responding to my comments.

Author Response

(The authors gave the same response as above.)
